# Effect of CaCO_3_ Particle Size on Surface Wetting and Adhesion: Studies on PMMA Model Substrates and *Laurus nobilis* Leaves

**DOI:** 10.3390/plants14243838

**Published:** 2025-12-17

**Authors:** Nora Mueller, Fabrizio Orlando, Victoria Fernandez, Gabriela Melo Rodriguez, Joachim Schoelkopf

**Affiliations:** 1Omya International AG, 4665 Egerkingen, Switzerland; 2Department of Biology, University of Fribourg, 1700 Fribourg, Switzerland; 3Systems and Natural Resources Department, School of Forest Engineering, Universidad Politécnica de Madrid, 28040 Madrid, Spain

**Keywords:** foliar application, calcium carbonate nanoparticles, wettability, surfactants

## Abstract

Leaf surfaces are protected by a hydrophobic cuticle with variable chemical composition and roughness, which often limits spray droplet retention and absorption. Optimizing foliar spray performance is therefore critical to maximize the desired effect on the target plant and minimize environmental impact. This study investigates the impact of particle size of calcium carbonate (CaCO_3_) in the presence and absence of a non-ionic surfactant on leaf surface deposition and wetting behavior. The tested formulations contained (i) no particles, (ii) CaCO_3_ nanoparticles, and (iii) CaCO_3_ microparticles (each at 2 wt%), applied using an airbrush or a handheld sprayer to polymethyl methacrylate (PMMA) plates, serving as model substrate, and on laurel leaves (*Laurus nobilis*). Water contact angle (WCA) measurements and coverage analysis were used to assess wetting performance. Initial WCA values were low (<12°) for all coatings, but rinsing revealed distinct behaviors. Coatings with nanoparticles retained a low WCA (<40°) and high coverage (>60%) after multiple rinsings, whereas microparticle coatings showed a sharp WCA increase (>60°) and significant coverage loss after few rinses. These findings demonstrate the long-lasting wetting effect of CaCO_3_ nanoparticles and highlight their potential as additives to enhance spray formulation performance.

## 1. Introduction

Foliar application of plant protection products, fertilizers or biostimulants is a routine technique for today’s agriculture in many areas of the world. Farmers rely on this application method to deliver nutrients, as it can be more target-oriented and environmentally friendly compared to root treatments [1]. Regarding plant protection products, the active ingredients must either be absorbed via a substantial plant surface area or be placed on the plant tissue of interest, depending on whether the mode of action is systemic or by contact. Estimations indicate that less than 0.1% of applied pesticides reach their target organisms [2], while most are lost due to run-off, spray drift, wash-off and photodegradation. Potential negative impacts of pesticides have been intensively studied [3], and there is a general consensus that risk reduction is crucial for protecting the environment. One approach to reach this goal is the improvement of leaf surface deposition efficacy [4]; however, the practical implementation of this approach is hindered by the fact that plant surface structure and composition vary considerably among plant species, varieties, organs or developmental stages [5], greatly affecting the uptake of nutrients through the leaves. In this context, leaf wettability, i.e., the ability of water or aqueous solutes to spread across or bead up on the surface of a leaf, is an essential factor to consider for effective foliar absorption. This property is measured by the contact angle (*θ*) of water droplets on the leaf surface. If the contact angle is less than 90°, the leaf is considered hydrophilic (water spreads easily). Conversely, if the contact angle is greater than 90°, the leaf is considered unwettable (water beads up); for example, micro-scale and nano-scale structural features can lead to super-hydrophobicity [6]. Barthlott and Neinhuis [7] first characterized the super-hydrophobic character of *Nelumbo nucrifera* leaves, commonly known as sacred lotus, resulting in a wave of research on superhydrophobic surfaces in nature and how to mimic them in technical applications [8]. Unfortunately for farmers, many crops, like cabbage or wheat, exhibit similar properties, resulting in treatment droplets having high contact angles and low roll-off angles [9,10]. Consequently, when dealing with such crops, foliar spray droplets not containing a surfactant will roll off the leaves, rendering the treatments ineffective for foliar uptake. The means to overcome this problem is to reduce the contact angles and increase the retention of spray drops by including surfactants, which are commonly used agricultural adjuvants that reduce surface tension [11,12]. The importance of leaf surface wettability may outweigh the contribution of factors associated with active ingredient physicochemical properties such as hygroscopicity [10,13]. When leaves of some species were treated with magnesium (Mg) or phosphorus (P) salts, with different degrees of hygroscopicity, a trend towards higher leaf uptake rates after the supply of hygroscopic compounds (i.e., having a low point of efflorescence and deliquescence) was only observed for unwettable maize (*Zea mais* L.), broccoli (*Brassica oleracea* L.) or leek (*Allium porrum* L.) leaves.

Regardless of plant species, it has been shown that good leaf coverage can be achieved in the field by optimizing droplet spreading and spray retention [11]. Droplet spreading pattern is influenced by factors such as nozzle type, droplet size and driving speed, while spray drop retention and drop shape (i.e., contact angles) can be modified by adding adjuvants into the tank mix [4]. However, despite the benefits of including adjuvants in spray formulations, they can also cause negative effects, like phytotoxicity [14] or impairment of leaf transpiration and photosynthesis [15].

The application of mineral element particle suspensions to agricultural crops, either as fertilizers, biostimulants or to reduce plant stress, is gaining popularity in agriculture. Indeed, particle films have been used for several plant protection purposes, such as pest control, disease control and beneficial effects on physiological and horticultural aspects [16]. For example, working with foliar-applied calcite and dolomite micro- and nano-sized particles, Pimentel et al. [17] observed not only the absorption of Mg and Ca but also a significant contribution to the uptake of soluble nutrient sources when used as foliar formulation adjuvants. In the context of climate change, particle films have been successfully used to reduce stress factors such as high leaf temperatures [18,19] and sunburn on fruit [20].

The previous study on foliar application of micro- and nano-sized CaCO_3_ particles [17] suggests that these particles may influence the formation of water film on leaves, thereby enhancing the rainfastness of foliar spray formulations. CaCO_3_ is abundant in nature, widely used as a liming agent, inexpensive, and not toxic [21], making it an attractive candidate for additional agricultural applications, superior to other nanoparticles [22,23]. Moreover, the use of nanoparticles in agriculture is gaining attention [24,25,26,27], with nano-CaCO_3_ showing promising effects on various crops in recent studies [28,29].

In this study we investigated the effect of CaCO_3_ particle size—comparing micro- and nano-particles—on surface wettability and coating stability. Using model PMMA substrates and laurel leaves (*Laurus nobilis*), we assessed water contact angle (WCA) evolution and coating retention after repeated rinsing steps simulating rain or dew, complemented by scanning electron microscopy (SEM) analysis. The findings offer insights into the mechanisms of particle adhesion and water interaction, providing a basis for optimizing foliar spray formulations to achieve improved efficacy and durability.

## 2. Results

Five foliar spray formulations were tested, differing in CaCO_3_ particle size and the presence or absence of surfactant (see Table 1 and Section 4.2 for details). The base formulation (BF_surf) contained surfactant but no mineral particles. Two formulations incorporated nanoparticles: NP (without surfactant) and NP_surf (with surfactant). The remaining two contained microparticles: MP (without surfactant) and MP_surf (with surfactant). All formulations were applied to rough PMMA plates and laurel leaves to assess wettability, retention, and coverage under rinsing conditions.

Figure 1 shows representative SEM images of the NP_surf (panels (a), (c), and (e)) and MP_surf coatings (panels (b) and (d)), as well as the bare PMMA plate (panel (f)), as illustrative examples. These images highlight significant differences in particle size and their impact on coating surface roughness. Specifically, CaCO_3_ nanoparticles form a dense and continuous surface layer, whereas the microparticles coating exhibits an irregular and discontinuous structure, resulting in increased surface roughness.

Figure 2 shows the water contact angle (WCA) measured on different coatings prepared with the formulations described in Section 4 and applied to a rough PMMA surface. The values correspond to freshly prepared coatings (solid bars) versus samples subjected to five rinsing steps (hatched bars). WCA was also measured on the bare PMMA surface for comparison (gray, patterned bar). The initial WCA is small for all the coatings, ranging from 4° to 12°, and therefore substantially lower than the ca. 83° measured on bare PMMA. The WCA increases upon rinsing, but to a different extent depending on the nature of the coating. Indeed, two distinct behaviors are observed for coatings with nanoparticles and microparticles. After five rinsing steps, the WCA of the coatings with nanoparticles increases slightly up to ca. 11° and 16° for samples with and without surfactant, respectively. Instead, the increase in WCA after rinsing is much more pronounced for the coatings containing microparticles, reaching values of 72° and 60° for samples with and without surfactant, respectively. The latter values resemble the WCA measured on the base formulation coating (without mineral particles) and compare well with the WCA measured on the bare PMMA plate.

Figure 3 shows the WCA data measured on different coatings after each rinsing step. The initial WCA, measured before rinsing (step nr. 0), is small for all substrates, but its evolution shows clear differences between coatings containing nanoparticles and microparticles. The largest WCA change upon rinsing is measured for the coatings with microparticles (blue and red dots) and with the base formulation (black squares), which is in line with the results presented in Figure 2. The WCA of these coatings increases to above 40° already after two rinsing steps. On the other hand, for the coatings containing nanoparticles, the WCA change is much lower. Specifically, the WCA measured on the coating with nanoparticles and surfactant (green dots) increases up to about 40° only after 25 rinsing steps, while the WCA measured on the coating without surfactant (yellow dots) only shows a minor increase from 7° to about 10°.

In line with the WCA measurements, SEM coverage analysis of the coatings on PMMA plates shows a clear difference related to the particle size of the mineral, as illustrated in Figure 4. The figure shows the coating coverage of MP_surf (a) and NP_surf (b) on PMMA substrates after 0, 5, and 26 rinsing steps. Microparticle coatings exhibited a sharp decline in coverage, dropping from approximately 53% initially to nearly 0% after five rinses, indicating poor retention. In contrast, nanoparticle coatings maintained substantially higher coverage, decreasing from approximately 77% to ca. 51% even after 26 rinses, which suggests superior adhesion and rainfastness compared to microparticles. Notably, statistical analysis confirms significant effect of particle size and rinsing steps on coating coverage across all rinsing condition. Representative SEM images of the as-prepared coatings and those after five rinsing steps for microparticles and nanoparticles are shown in Figure 4c,e and Figure 4d,f, respectively. The images reveal that coverage achieved with the MP_surf formulation is noticeably lower compared to that of the NP_surf formulation.

Figure 5 shows the coverage of coatings applied on laurel leaves before and after five rinsing steps (the preparation procedure and analysis are described in Section 4). In line with the results presented in Figure 4, nanoparticles-based coatings exhibited greater initial coverage and superior retention after rinsing compared to those formulated with microparticles, confirming their enhanced adhesion and durability on natural leaf surfaces. Statistical analysis confirms significant coverage differences between microparticle and nanoparticle coatings on laurel leaves, although the rinsing procedure did not produce a statistically significant effect in this case compared with the effect observed on PMMA.

## 3. Discussion

Nowadays, foliar sprays containing mineral particles such as kaolinite or CaCO_3_ are increasingly applied to agricultural crops to reduce insect or pathogen damage [30,31] or to improve plant tolerance to excess heat or irradiation stress [32,33]. A variety of commercial products based on mineral particles of different sizes are already available, but their mode of action and plant responses to the treatments remain unclear. Some such natural products, which contain nutrient elements like CaCO_3_ may also act as slow-release fertilizers, provided they are retained on the foliage after spraying [17].

Hence, we focused on evaluating the influence of CaCO_3_ microparticles and nanoparticles on surface wettability and water retention using a gradual experimental approach that included trials on both model and plant surfaces. CaCO_3_ nanoparticles are particularly interesting compared to other nanomaterials because CaCO_3_ is abundant in nature, widely used as a liming agent, inexpensive, and not toxic [21]. These properties largely eliminate the risk of harmful environmental accumulation, even at high application rates or with repeated use, and make them particularly suitable for leaf applications in contrast to other nanoparticles [22,23].

The findings reveal that, depending on surface nature and particle size, CaCO_3_ was either retained or removed after washing with water. This washing step was introduced to emulate the effect of rain or dew on spray drop deposits on leaves, as CaCO_3_ particles have shown great potential, at least as adjuvants, in foliar fertilizer sprays [17]. Initial trials were conducted on PMMA plates as a model surface to ensure work on homogenous, standardized and well-characterized substrates, which is difficult to achieve with leaves. Additionally, plant leaves exhibit highly diverse morphology and surface properties, meaning that results on specific leaves may not be universally applicable. This study setup aims to clarify the impact of CaCO_3_ particle size while acknowledging that outcomes in whole-plant and field studies may differ significantly.

WCAs of coated PMMA surfaces, prepared with the base formulation with and without particles, were measured on freshly prepared coatings. This initial measurement showed WCA values lower than 12° on all surfaces. Accordingly, the low contact angle initially measured on the coatings can be attributed to their specific surface chemistry, whereas the low contact angle after rinsing the surfaces can be related to their roughness. Several studies have reported nanoparticles may modify surface roughness, enhancing various functional properties [34,35,36]. This indicates that the surfactant used in the coatings increased the spreading of water droplets by reducing surface tension, which is in line with the low WCA values observed (Figure 2). On the other hand, the situation changed dramatically upon rinsing the treated surfaces: nanoparticle formulations showed a small increase in WCA, in contrast to the large WCA increase measured on the other coatings, suggesting that particle size influences wettability independently of the surfactant presence. We attribute these observations to the effect of liquid menisci forming around the particles, be it either from the evaporating formulation liquid or from precipitation. The curvature of these menisci—dependent on the contact angle *θ* and characterized by radius *r_m_* (Figure 6)—leads to a Laplace pressure *P_c_*, which in this case represents the pressure difference across a curved liquid-vapor interface, where *γ* is the interfacial surface tension. The higher pressure is always on the convex side [37].
(1)$$P_{c}=\frac{2\gamma }{r_{m}}.$$

The Laplace pressure relates to the vapor pressure through the Kelvin equation:
(2)$$P_{vc}=P_{v0}\exp\left(P_{c}k\right)$$ where *P_vc_* is the vapor pressure above the curved meniscus, *P_v_*_0_ is the vapor pressure above a flat liquid surface, and *k* = *Vm*/*RT*, with *V_m_* being the molecular volume, *R* the gas constant, and *T* the temperature. It is evident from Equation (2) that the concave liquid meniscus reduces evaporation, in contrast to the convex meniscus of a droplet. Furthermore, the smaller the particles, the greater their number per unit weight, the more menisci are formed, and the stronger the overall effect. As illustrated in Figure 6 for the case of spherical particles, the thinner the liquid film becomes through evaporation, the stronger the meniscus curvature and the stronger the Laplace pressure become. Another consequence of the Laplace pressure around the particles is that the menisci trigger strong adhesion to the surface.

The results in Figure 3 indicate that WCA values for formulations containing CaCO_3_ nanoparticles remain low, even after multiple rinsing steps. This prolonged effect is not observed for the other samples, suggesting that the surfactant effect progressively fades away with each rinsing step, eventually disappearing after five rinsing steps. Moreover, the surface coverage of coatings containing microparticles drops dramatically after five rinsing steps (Figure 4). Taken together, the results suggest that low WCAs are associated with the presence of CaCO_3_ nanoparticles on the surfaces. We tentatively attribute this to an interplay between particle size and surface roughness. Nanoparticles with a d50 of 91 nm can deposit and accumulate in the valleys of the micro-rough PMMA surface, where they remain trapped. This process is less likely for CaCO_3_ microparticles, as they have a diameter similar to the characteristic roughness of the surface, preventing effective deposition or accumulation in the valleys of the coarse PMMA substrate. Therefore, rinsing the surface with water may be less effective for the coatings containing nanoparticles, since these smaller particles are more difficult to remove. The slightly lower coverage observed for the formulation with surfactant may be explained by the surfactant reducing the adhesion of nanoparticles to the PMMA plate. The SEM micrographs shown in Figure 4 confirm that the microparticle coating not only results in a lower absolute coverage but is also less resistant against removal by water washing compared to coating with nanoparticles.

The dramatic drop in surface coverage observed for CaCO_3_ microparticle coatings on PMMA plates was less pronounced on laurel leaves (Figure 5). This may be attributed to chemical and structural differences between PMMA and laurel leaf surfaces, including the potential loss of surface structure due to leaf dehydration. In contrast, the nanoparticle coverage remained nearly unchanged on laurel leaves after five rinsing steps, confirming the stronger adhesion of smaller particles to leaf surfaces, as reported in some investigations [38,39].

In general, leaves coated with nanoparticles exhibited more uniform coverage and stronger particle retention compared to those coated with microparticles. The enhanced wetting and retention properties provided by mineral particles, chiefly those of nano size, make them promising candidates for use as additives in agrochemical spray formulations [17].

When considering potential surface interactions between CaCO_3_ particles and leaf surfaces, additional physio-chemical properties—like the surface free energy of the particles themselves—may play a critical role. For example, the hydrophilic nature of CaCO_3_ in terms of acid-base interactions [40] and the reorganization of the nanoparticles provided as an aqueous suspension on PMMA or leaf surfaces may play a role that should be elucidated in future investigations. The sum of such effects will ultimately dominate the degree of surface coverage and retention for the applied mineral microparticles and nanoparticles. The reorganization of the nanoparticles can happen after the evaporation of a liquid that is in contact with the surface; an example of this effect is the coffee ring, where the nanoparticles accumulate, creating a ring [41] and making a path for the fluid that will be in contact in the future [42,43].

## 4. Materials and Methods

### 4.1. Microparticles and Nanoparticle Characterization

CaCO_3_ microparticles, supplied by Omya AG, had a volume median diameter, d50, of 3.5 μm and a d98 of 9.9 μm. The detailed particle size distribution curve of the microparticles was measured by sedimentation technique (Sedigraph, Micromeritics, Norcross, GA, USA) and is reported in Appendix A. Nanoparticles were obtained by bead mill grinding (MicroMedia P1, Bühler, Uzwil, Switzerland) of these microparticles. A CaCO_3_ slurry with 10% solids content (SC) was prepared and sieved using a mesh with 100 µm openings. The bead mill was then circulated with the prepared slurry and operated at a rotor speed of 1300 rpm for approximately 30 min, with a slurry feed rate of 50% pump speed. Zirconia beads of 0.1 mm diameter were used. The particle size of the CaCO_3_ in suspension was regularly checked during the grinding process. To prevent agglomeration caused by their poor stability in aqueous medium, the nanoparticles were freeze-dried after the grinding step. The particle size distribution of the nanoparticles was measured by dynamic light scattering (Zetasizer Nano ZS, Malvern, Malvern, UK), while specific surface area was determined using the BET (Brunauer-Emmet-Teller) method using nitrogen as adsorbing gas (ASAP 2460, Micromeritics). The resulting nanoparticles exhibited a number-based median diameter d50 of 91 nm and a specific surface area of 38 m^2^/g, qualifying as a nanomaterial according to the EU definition (2011/696/EU). The detailed particle size distribution curve of the nanoparticles is shown in Appendix A.

### 4.2. Foliar Spray Formulations

The water-based formulations were prepared by adding 5 g of mineral powder to 245 g of CaCl_2_ solution (150 mM) and, in some cases, 0.25 g of Break-Thru (BT) (AlzChem Group AG, Trostberg, Germany) in deionized water. A base formulation without mineral particles was also prepared. The composition of these foliar spray formulations is detailed in Table 1.

### 4.3. Coating of PMMA Plates and Laurel Leaves

Uniform mineral coatings were obtained by spraying the foliar formulations onto PMMA plates with a roughness parameter S_a_ of approximately 5 μm (WW5, Schönberg, Hamburg, Germany). The formulations were sprayed using an airbrush (SP-575, Sparmax, Taipei, Taiwan) with the nozzle placed 30 cm above the center of the PMMA plate. Each spray lasted 5 s, followed by a gentle drying under an IR lamp for approximately 5 min before the next layer was applied upon a quarter rotation of the plate to ensure homogeneity; the procedure was repeated until a coating weight of ca. 15 mg was achieved.

Laurel leaves were harvested shortly before the application of the mineral coating and placed on a flat surface. The same formulations above were applied with a handheld sprayer (Super Star 1.25/360°, Birchmeier, Stetten, Switzerland) at a 45-degree angle at approximately 30 cm distance. Each leaf was sprayed three times from different sides, at the same angle and distance. The second and third layers of suspension were applied when water was evaporated on the leaf surface. After the last application, the leaves were dried for 30 min and then gently covered with a glass plate to avoid curling.

### 4.4. Rinsing Procedure

Particle-coated PMMA plates or laurel leaves were placed on a sample holder at a 45° angle and positioned below a bucket at a distance of 45 cm. The bottom of the bucket was perforated beforehand with 1 mm diameter holes to simulate raindrops. To control the rinsing procedure, a mount for a plate was fixed between the bucket and the samples. The bucket was filled with tap water, and the plate was removed once the water level reached the start mark. Start and stop levels were calculated by taking into account the decreasing flow rate, which correlates with the water level. The amount of water was set to 1360 mL per rinsing step. Between rinsing steps, coated PMMA plates were stored under controlled conditions (23 °C, 50% relative humidity (RH)) for at least 1 h. Laurel leaves were kept in the lab in a slightly tilted position to reduce drying time without causing fast runoff of the residual wash water. The unregulated runoff of water could potentially lead to the formation of new structures in the particle distribution. On a plant, runoff water from higher-located leaves complicates keeping the reproducible rinsing volume and could also add random mineral particles to leaves below. Therefore, the leaves used in this study were detached from the plant.

### 4.5. Scanning Electron Microscopy (SEM)

Micrographs were taken with a backscattered electron detector (NTS BSD) mounted on an SEM (Zeiss EVO LS 15, Zeiss, Oberkochen, Germany). By running the detector in COMPO mode, it is possible to visualize qualitative differences in the elemental composition of a sample. The heavier the atomic weight of the present element, the brighter the particle appears in the image, allowing areas where the mineral coating is present to be identified.

### 4.6. Contact Angle Measurements

Water contact angle (WCA) measurements were conducted with an optical contact angle meter (OCA 50; DataPhysics, Filderstadt, Germany) by analyzing the contour of water drops with SCA-20 software (Version 4.5.13), using the circle and ellipse method. The dosing system consisted of a syringe filled with deionized water, which had a 0.52 mm external diameter needle for dispensing small volume drops. The coated samples were pre-conditioned for at least 24 h before measurement at 23 °C and 50% RH. For the measurements, the coated surface of interest was located under the dosing system, and 2 µL drops of deionized water were deposited on it for contact angle analysis. The average contact angle (left and right contact angle) was measured using the methods of circle (less than 30°) or ellipse fitting (between 30° and 60°).

### 4.7. Coverage Analysis

To assess coating coverage on PMMA plates, a systematic image acquisition approach was applied using SEM at 50× magnification. A 5 × 5 grid of 25 images was captured, each with a pixel size of 0.744 µm, resulting in image dimensions of 2285 × 1714 µm. The images were spaced to ensure coverage of a representative sample area, resulting in an analyzed region of approximately 29.7 × 22.3 mm. These image arrays were then processed to quantify coating distribution and assess uniformity across the selected regions. The image arrays are shown in Appendix A.

Coating coverage on laurel leaves was quantified by image analysis. Only coatings prepared with MP_surf and NP_surf formulations were considered. For each formulation, an image containing four coated leaves and one uncoated leaf was converted to grayscale and segmented using a global intensity threshold. Threshold values were determined from the uncoated leaf to ensure accurate discrimination of coated regions. Coated and uncoated areas were assigned a value of 1 and 0, respectively. Coverage was calculated as the ratio of coated to total pixels. Reported values represent the mean of the four coated leaves, and the standard deviation was calculated from these replicates.

For both PMMA and laurel leaf cases, mineral coating coverage was analyzed via one-way analysis of variance (ANOVA). When ANOVA indicated a significant effect (*p* < 0.05), pairwise comparisons were conducted using Tukey’s HSD test.

## 5. Conclusions

Foliar formulations containing CaCO_3_, especially in the form of nanoparticles, enhance a long-lasting wetting effect on applied surfaces. It was shown that the water contact angle on surfaces with formulations containing nanoparticles with and without surfactant is below 40°, even after exposing the samples to multiple rain cycles. The SEM mapping analysis also showed how the CaCO_3_ nanoparticles remain on the surfaces after the rain cycles. These results, together with the inspected observations, corroborated that water layers were retained on the surface, which is necessary for nutrient mobility; we suggest that this is due to the water menisci forming around the particles. This effect would also explain the lower contact angles observed when the coating contains the sub-micron particles. At the triple line, i.e., the three-phase confluence line of the WCA droplet, we suppose the particles give rise to a “jagged” wetting line triggered by the local Laplace pressures between and around the particles pulling the droplet boundary outwards to exhibit a lower contact angle than without particles [34,35,36].

This study suggests that CaCO_3_ can improve the surface interaction of foliar-applied agrochemical formulations. Nonetheless, the variability in leaf morphology and surface properties across plant species may influence these interactions, limiting the universal applicability of our findings. Additionally, environmental conditions could affect the performance of CaCO_3_ nanoparticles under whole-plant and field scenarios. Despite these considerations, the findings highlight promising opportunities for the use of CaCO_3_ in foliar treatments, which should be evaluated in future studies.

## Figures and Tables

**Figure 1 plants-14-03838-f001:**
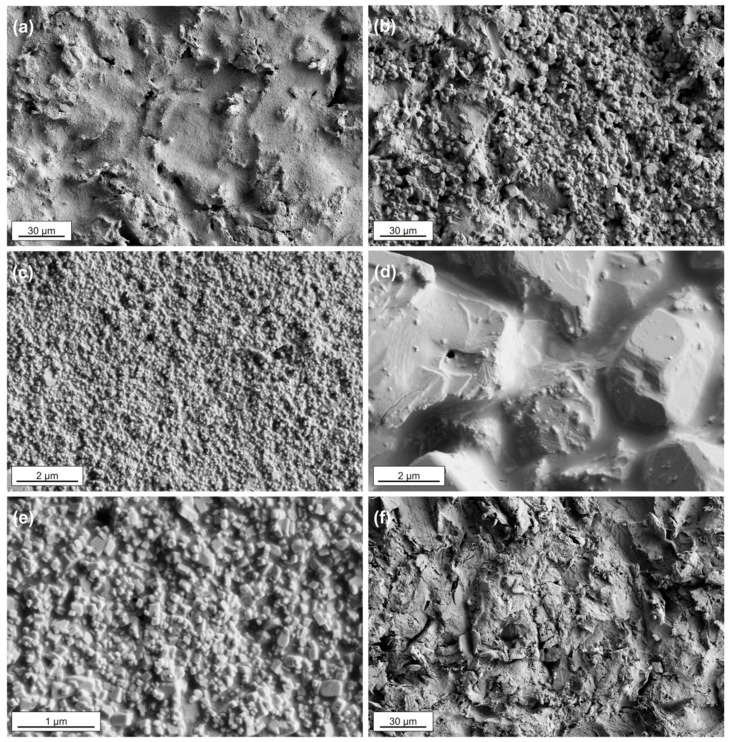
SEM images showing the surface morphology of mineral coatings applied on PMMA plates at different magnifications. Panels (**a**,**c**,**e**) correspond to the NP_suf coating; panels (**b**,**d**) show the MP_surf coating; panel (**f**) displays the bare PMMA model substrate.

**Figure 2 plants-14-03838-f002:**
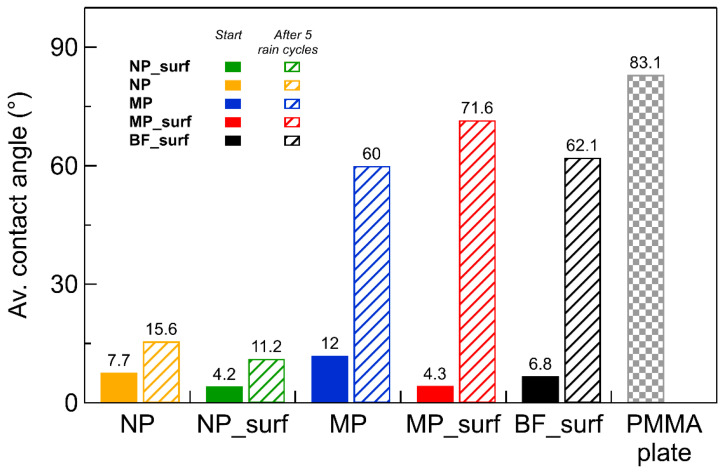
Average water contact angle measured on the different coatings realized with formulations containing nanoparticles with or without surfactant (NP_surf and NP, respectively), microparticles with or without surfactant (MP_surf and MP, respectively), and no mineral particles (BF_surf). The water contact angle was measured also on the raw surface of the baking PMMA plate (checked bar). Solid and hatched bars refer to the values before and after application of five rinsing steps, respectively.

**Figure 3 plants-14-03838-f003:**
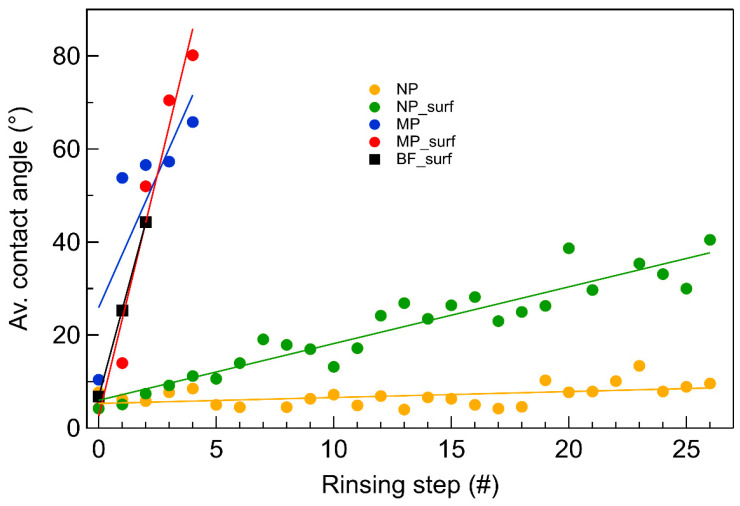
Evolution of the average water contact angle as a function of the number of applied rinsing steps. The measurements were performed on the different coatings realized with formulations containing nanoparticles with or without surfactant (NP_surf and NP, respectively), microparticles with or without surfactant (MP_surf and MP, respectively), and no mineral particles (BF_surf).

**Figure 4 plants-14-03838-f004:**
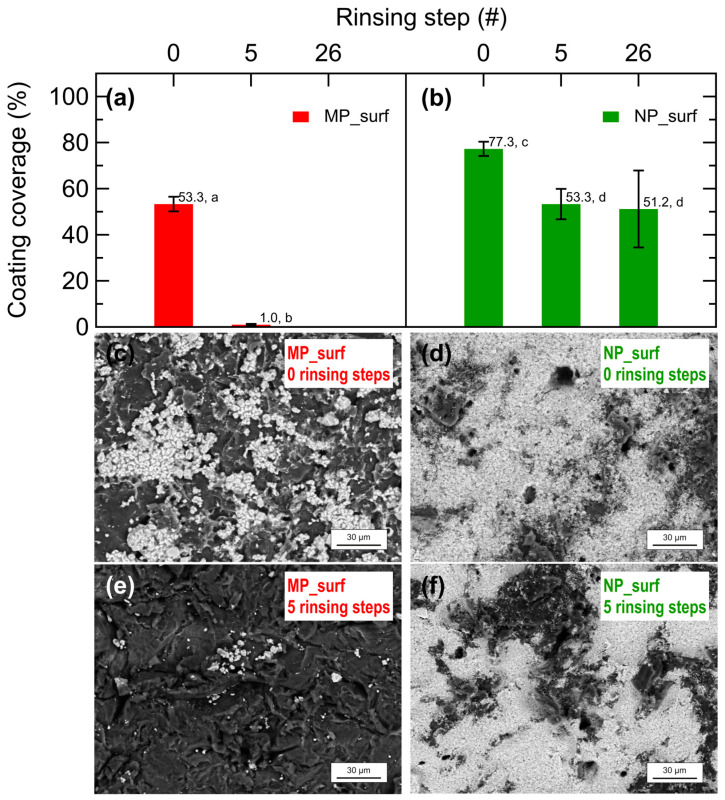
Evolution of coating coverage on PMMA substrate as a function of rinsing steps for (**a**) MP_surf and (**b**) NP_surf mineral coatings. Data correspond to mean coverage and error bars represent standard deviation. Different letters indicate homogenous groups according to Tukey’s HSD test (*p* < 0.05). Panels (**c**,**e**) show SEM images of the MP_surf coating in the as-prepared state and after five rinsing steps, respectively. Panels (**d**,**f**) show SEM images of the NP_surf coating in the as-prepared and after five rinsing steps, respectively.

**Figure 5 plants-14-03838-f005:**
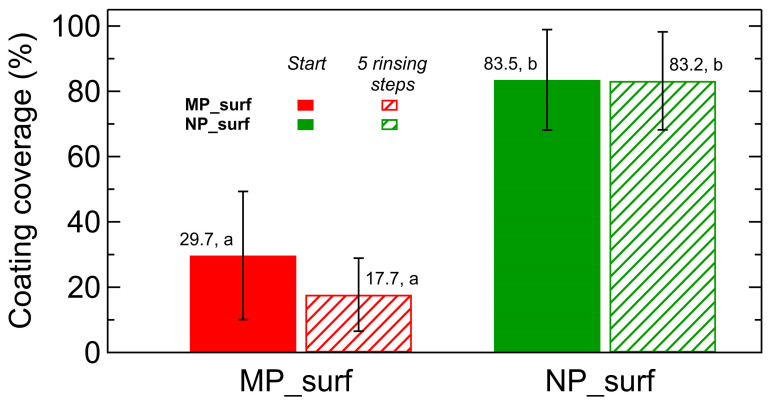
Coating coverage on *Laurus nobilis* leaves for MP_surf (red) and NP_surf (green) mineral coatings. Solid bars represent the as-prepared state, and hatched bars represent coatings after five rinsing steps. Data correspond to mean coverage and error bars indicate standard deviation. Different letters indicate homogenous groups according to Tukey’s HSD test (*p* < 0.05).

**Figure 6 plants-14-03838-f006:**
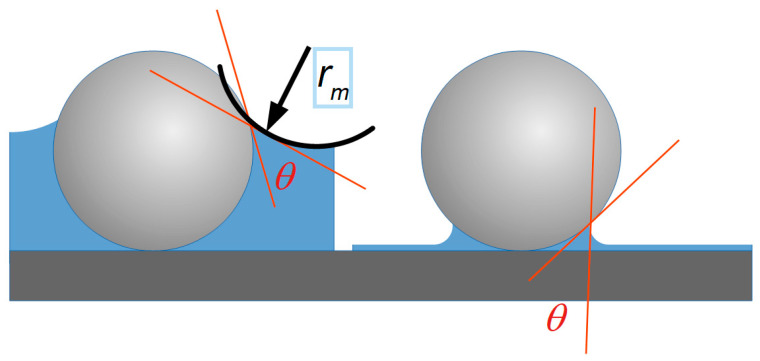
Schematic representation of particles (light gray) on a surface (dark gray) in the presence of water (blue): on the left, as a thicker film; on the right, as the film approaches rupture through evaporation. The WCA (*θ*) is shown as approximately 30°, and the radius of the meniscus curvature (*r_m_*) is indicated in the left case.

**Table 1 plants-14-03838-t001:** Aqueous suspensions formulation for each coating.

ID	Nano CaCO_3_ (d50: 135 nm)	Micro CaCO_3_	Surfactant	CaCl_2_ 150 mM
	(g)	(g)	(g)	(g)
Base Formulation (BF_surf)	-	-	0.25	245.0
Nanoparticle (NP)	5.0	-	-	245.0
Nanoparticle + Surfactant (NP_surf)	5.0	-	0.25	245.0
Microparticle (MP)	-	5.0	-	245.0
Microparticle + Surfactant (MP_surf)	-	5.0	0.25	245.0

## Data Availability

The original contributions presented in this study are included in the article/Appendix A. Further inquiries can be directed to the corresponding authors.

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
