# Peer review of "Effect of CaCO_3_ Particle Size on Surface Wetting and Adhesion: Studies on PMMA Model Substrates and *Laurus nobilis* Leaves"

_plants, 2025, doi:10.3390/plants14243838_

Round 1

Reviewer 1 Report

Comments and Suggestions for Authors

The manuscript addresses an interesting and relevant topic with potential scientific value; however, in its current form, it requires substantial revisions before it can be considered for publication. Several aspects of the experimental design, data presentation, and characterization methods need to be clarified or expanded. In particular, high resolution SEM images of leaves can add to the support of the conclusions, and key characterization data for the synthesized nanoparticles (e.g., DLS size distribution, BET surface area, and direct imaging of nanoparticles) are missing. Furthermore, certain terminological  inconsistencies should be resolved in accordance with accepted standards (e.g., ISO and IUPAC definitions of nanoparticles).

The section on nanoparticle production (Lines 249 to 262: 4.1. Nanoparticle production and characterization) and characterization requires additional data and clarification to substantiate the claims. Although the particle size was reportedly determined by DLS and the surface area by BET analysis, the corresponding results (e.g., DLS size distribution curve and BET surface area values) are not presented. These datasets should be included to enable proper evaluation of the nanoparticle size distribution, homogeneity, and specific surface characteristics.

In addition, the reported median particle size (d₅₀ = 135 nm) exceeds the widely accepted upper limit for nanoparticles. According to both the International Organization for Standardization (ISO, ISO/TS 80004-1:2015) and the International Union of Pure and Applied Chemistry (IUPAC) recommendations, nanoparticles are defined as particles with all external dimensions in the nanoscale, typically between 1 and 100 nanometres. Particles with dimensions above 100 nm should therefore be described as submicron particles or nanostructured materials, rather than true nanoparticles. The terminology used in the manuscript should be revised accordingly, or the size data should be verified to confirm whether the produced material indeed meets the nanoscale definition.

Finally, while SEM was employed to visualize changes on the leaf surface, the manuscript does not include any SEM or TEM images of the synthesized CaCO₃ nanoparticles themselves. Providing such images is essential to confirm their morphology and size, and would greatly strengthen the validity of the nanoparticle characterization. It may even provide proof that the created particles can be defined as nanoparticles, since DLS measurement take into account the corona made of water molecules moving with nanoparticles. I expect SEM or TEM images to show particles with median size around 90 to 115 nm.

The SEM analysis sections (Lines 124 to 137 and lines 300 to 309) describes the acquisition of numerous micrographs of leaf surfaces; however, all presented images were taken at relatively low magnification (200×). At this scale, it is not possible to directly observe individual nanoparticles. Given the capabilities of the SEM instrument described (Zeiss EVO LS 15) and the use of backscattered electron detection, it should be feasible to obtain higher-magnification images that clearly resolve the deposited nanoparticles or nanoparticle aggregates. That can provide invaluable insight on how they are deposited on the leaf surface, not just the coverage area. I therefore recommend including representative high-resolution SEM images (close-ups) showing the nanoparticles on the leaf surface to provide direct visual evidence of their deposition and interaction with the tissue.

Author Response

The manuscript addresses an interesting and relevant topic with potential scientific value; however, in its current form, it requires substantial revisions before it can be considered for publication. Several aspects of the experimental design, data presentation, and characterization methods need to be clarified or expanded. In particular, high resolution SEM images of leaves can add to the support of the conclusions, and key characterization data for the synthesized nanoparticles (e.g., DLS size distribution, BET surface area, and direct imaging of nanoparticles) are missing. Furthermore, certain terminological  inconsistencies should be resolved in accordance with accepted standards (e.g., ISO and IUPAC definitions of nanoparticles).

We thank the Reviewer for the overall assessment and constructive feedback. The points raised regarding experimental design, data presentation, characterization methods, and terminology have been addressed in detail in the corresponding responses below.

The section on nanoparticle production (Lines 249 to 262: 4.1. Nanoparticle production and characterization) and characterization requires additional data and clarification to substantiate the claims. Although the particle size was reportedly determined by DLS and the surface area by BET analysis, the corresponding results (e.g., DLS size distribution curve and BET surface area values) are not presented. These datasets should be included to enable proper evaluation of the nanoparticle size distribution, homogeneity, and specific surface characteristics.

We thank the Reviewer for this valuable comment and agree that providing detailed data will improve the manuscript. In response, we have:

  • created a Supplementary Materials document and added the DLS distribution curve to illustrate nanoparticle size and homogeneity (Figure S2). For completeness we also included the particle size distribution of the microparticles measured with the sedimentation technique (Figure S1).
  • Included the BET surface area value in Section 4.1.

These additions are now referenced in the revised manuscript for clarity. We believe these additions address the Reviewer’s concern and enable a more comprehensive evaluation of nanoparticle production and characterization.

In addition, the reported median particle size (d₅₀ = 135 nm) exceeds the widely accepted upper limit for nanoparticles. According to both the International Organization for Standardization (ISO, ISO/TS 80004-1:2015) and the International Union of Pure and Applied Chemistry (IUPAC) recommendations, nanoparticles are defined as particles with all external dimensions in the nanoscale, typically between 1 and 100 nanometers. Particles with dimensions above 100 nm should therefore be described as submicron particles or nanostructured materials, rather than true nanoparticles. The terminology used in the manuscript should be revised accordingly, or the size data should be verified to confirm whether the produced material indeed meets the nanoscale definition.

We thank the Reviewer for this constructive comment, which allowed us to clarify an important point. The reported d50 of 135 nm in the submitted manuscript refers to a volumetric distribution, which allows direct comparison with the microparticles, whose d50 value of 3.5 μm is also based on a volumetric basis.

When assessed on a number basis, the particle size of the nanoparticles shows a d50 of 95 nm, meaning that 50% of the particles are below 100 nm. In line with the European Commission recommendation (2011/696/EU), a material is considered a nanomaterial if 50% or more of the particles (by number) have at least one external dimension below 100 nm. Therefore, our material qualifies as a nanomaterial under this definition.

Regarding the ISO/TS 80004-1:2015 standard, we acknowledge that its purpose is to harmonize terminology rather than establishing regulatory thresholds. It does not specify a numerical criterion for classification, whereas the EU recommendation provides a clear quantitative threshold for regulatory purposes.

Based on the above, and to avoid confusion, we revised the manuscript to clarify that the classification is based on the EU definition and updated the d50 value of the nanoparticles to reflect a number-based distribution.

In addition, we have included details about the particle size distribution for the nanoparticles, and the microparticles, in the newly created Supplementary Materials.

Finally, while SEM was employed to visualize changes on the leaf surface, the manuscript does not include any SEM or TEM images of the synthesized CaCO₃ nanoparticles themselves. Providing such images is essential to confirm their morphology and size, and would greatly strengthen the validity of the nanoparticle characterization. It may even provide proof that the created particles can be defined as nanoparticles, since DLS measurement take into account the corona made of water molecules moving with nanoparticles. I expect SEM or TEM images to show particles with median size around 90 to 115 nm.

We thank the Reviewer for this valuable comment. Unfortunately, SEM was not employed to image the surface of coated leaves due to technical constraints related to preserving surface integrity. Although SEM is commonly used to examine leaf morphology, samples must withstand demanding vacuum conditions and extensive preparation steps, typically including chemical fixation, alcohol dehydration, critical point drying, and sputter coating, or alternatively, cryo-fixation. Both approaches, however, are prone to introducing artefacts such as leaf shrinkage under low vacuum and cracking of cryo-fixed cell surfaces, which could cause the coating to crack or detach from the leaves and thereby compromise or alter the surface structure and its evaluation.

We have now included SEM images of both CaCO₃ microparticles and nanoparticles coatings deposited on a PMMA plate in Figure 1 of the revised manuscript. These images confirm that the nanoparticles are below 100 nm, supporting their classification as nanomaterials. While an SEM image analysis to extract a particle size distribution was not performed, the visual evidence complements the DLS and BET data presented in the manuscript and Supplementary Materials.

The SEM analysis sections (Lines 124 to 137 and lines 300 to 309) describes the acquisition of numerous micrographs of leaf surfaces; however, all presented images were taken at relatively low magnification (200×). At this scale, it is not possible to directly observe individual nanoparticles. Given the capabilities of the SEM instrument described (Zeiss EVO LS 15) and the use of backscattered electron detection, it should be feasible to obtain higher-magnification images that clearly resolve the deposited nanoparticles or nanoparticle aggregates. That can provide invaluable insight on how they are deposited on the leaf surface, not just the coverage area. I therefore recommend including representative high-resolution SEM images (close-ups) showing the nanoparticles on the leaf surface to provide direct visual evidence of their deposition and interaction with the tissue.

We would like to clarify that SEM imaging was not performed on the coated leaf surfaces for technical reasons explained above. The sections cited by the Reviewer (Lines 124–137 and 300–309) refer specifically to SEM images acquired on PMMA plates, which were used as a model substrate for particle deposition studies.

We acknowledge the importance highlighted by the Reviewer of high-resolution imaging for visualizing individual nanoparticles on leaf surfaces. However, due to the inherent complexity and variability of biological samples, achieving reliable high-magnification SEM imaging on coated leaves was not feasible within the scope of this work. To ensure reproducibility and accurate interpretation of particle distribution, our approach focused on PMMA plates.

In response to the Reviewer’s comment, we replaced the SEM images of mineral coatings with higher-resolution ones in Figure 3 of the submitted manuscript, which correspond to Figure 4 of the revised version. Additionally, the low-resolution SEM images used for evaluating the coverage have been included in the Supplementary Materials (Figure S3–S6).

Reviewer 2 Report

Comments and Suggestions for Authors

Comment 01:

The manuscript titled " Calcium Carbonate Particles as Wetting Booster for Foliar Application in Agriculture" (ID: plants-3961364) presents an innovative approach to enhancing the effectiveness of foliar sprays, a crucial method in plant nutrition . the study investigates the potential of sub-micron calcium carbonate (CaCO3) nanoparticles, combined with surfactants, to improve wettability and retention of spray droplets on plant surfaces, specifically targeting the challenges posed by the hydrophobic nature of plant cuticles.  The findings highlight the superior performance of CaCO3  nanoparticles over micro-particles, showing a long-lasting wetting effect and potential for enhancing spray retention, even after multiple rinsing steps. This work is highly relevant to the agricultural sector, offering promising implications for optimizing the application of fertilizers, plant protection products, and biostimulants, which could lead to more eficient agricultural practices and improved crop yields. The socio economic impact of this research could be significant, particularly in improving resource use efficiency, reducing environmental impacts, and enhancing crop productivity in the face of increasing global agricultural demands .

Comment 02:

however, I have several critiques regarding this manuscript. Firstly, the title appears to be rather vague and could be more suited for a review article rather than a research paper. The phrase "Foliar Application in Agriculture" is  broad and lacks specificity, especially considering the study focuses on the application of calcium carbonate nanoparticles on Laurus leaves. The title does not adequately reflect the pecificity of the experimental work conducted, which could lead to ambiguity about the manuscript’s scope .

Comment 03:

The  abstract lacks sufficient detail regarding the experimental design, which makes it difficult  to fully assess the robustness of the study. While it outlines the general approach of investigating the impact of calcium carbonate nano particles on wettability and retention, it does not provide enough information on the experimental conditions, such as concentrations of  nanoparticles used, the specific surfactants tested , or the precise methods of aplication and measurement .

Comment 04:

Additionally  the abstract does not include any quantitative data or specific numerical results, such as the actual contact angle measurements or the extent of spray  retention improvements .

Comment 05:

The manuscript would benefit from incorporating more recent references, particularly in the introduction.   currently, it cites only one reference from 2024 and none from 2025, which is unusual for a paper submitted in late 2025 .

Comment 06:

The statistical analyses performed in this study appear to be basic and lack sufficient depth. There are no comparisons of means, suchANOVA , to statistically validate the differences observed between the various experimental conditions.

Comment 07:

The figure and table legends should be more self-contained. Although the information may be presented in the body of the text, it is crucial that all relevant details are included in the legends for clarity and ease of understanding. This includes the number of repetitions in the experiments, the nature of the error values (standard deviation or standard error.....), and explanations of any abbreviations used within the figures or tables .

Comment 08:

  • Please, make sure that all references have a corresponding citation within the text and vice versa.
  • Please, double-check the spelling of the author’s names and dates and make sure they are correct and consistent with the citations.
  • Please, spel out all journal titles in the references section.
  • Please, make sure that all figures and tables are cited within the text and that they are cited in consecutive order.
  • Please, spell al the abbreviations the first time when they are mentioned in the text.

Comment 09:

I am sorry, but the article is far from meeting the standards of the Plants journal. The manuscript requires significant revisions, particularly i n terms of experimental design, statistical analysis, and the inclusion of more recent reference s.

Author Response

Comment 01:

The manuscript titled " Calcium Carbonate Particles as Wetting Booster for Foliar Application in Agriculture" (ID: plants-3961364) presents an innovative approach to enhancing the effectiveness of foliar sprays, a crucial method in plant nutrition. The study investigates the potential of sub-micron calcium carbonate (CaCO3) nanoparticles, combined with surfactants, to improve wettability and retention of spray droplets on plant surfaces, specifically targeting the challenges posed by the hydrophobic nature of plant cuticles.  The findings highlight the superior performance of CaCO3  nanoparticles over micro-particles, showing a long-lasting wetting effect and potential for enhancing spray retention, even after multiple rinsing steps. This work is highly relevant to the agricultural sector, offering promising implications for optimizing the application of fertilizers, plant protection products, and biostimulants, which could lead to more efficient agricultural practices and improved crop yields. The socio-economic impact of this research could be significant, particularly in improving resource use efficiency, reducing environmental impacts, and enhancing crop productivity in the face of increasing global agricultural demands.

We appreciate the recognition of the manuscript’s relevance to agricultural practices. We are pleased that the Reviewer acknowledges the innovative approach and the potential socio-economic impact of this research, and we thank the Reviewer for highlighting the significance of our findings in improving resource efficiency and crop productivity. 

Comment 02:

However, I have several critiques regarding this manuscript. Firstly, the title appears to be rather vague and could be more suited for a review article rather than a research paper. The phrase "Foliar Application in Agriculture" is broad and lacks specificity, especially considering the study focuses on the application of calcium carbonate nanoparticles on Laurus leaves. The title does not adequately reflect the specificity of the experimental work conducted, which could lead to ambiguity about the manuscript’s scope.

We thank the Reviewer for the valuable feedback regarding the manuscript title. In response, we have changed the title to “Effect of CaCO3 particle size on surface wetting and adhesion: Studies on PMMA model substrates and Laurus nobilis leaves” to provide greater specificity and clarity about the scope of our work. This revised title more accurately reflects the experimental focus, emphasizing that our study investigates the effect of particle size—including both nano- and micro-particles—on wetting and adhesion, rather than exclusively targeting nanoparticles. We believe this change better aligns the title with the content and objectives of the manuscript.

Comment 03:

The abstract lacks sufficient detail regarding the experimental design, which makes it difficult to fully assess the robustness of the study. While it outlines the general approach of investigating the impact of calcium carbonate nano particles on wettability and retention, it does not provide enough information on the experimental conditions, such as concentrations of nanoparticles used, the specific surfactants tested , or the precise methods of application and measurement .

We acknowledge that including key experimental details in the abstract would improve clarity. In the revised version, we have added information on the concentrations of calcium carbonate nanoparticles and microparticles (2 wt%), the surfactant tested, and the methods used for application (Airbrush application and handheld sprayer) and measurement (WCA and image analysis) to ensure the experimental design is more transparent and robust.

Comment 04:

Additionally the abstract does not include any quantitative data or specific numerical results, such as the actual contact angle measurements or the extent of spray retention improvements .

We appreciate the reviewer’s feedback regarding the inclusion of quantitative data and specific numerical results in the abstract. In the revised abstract, we have now included the results of contact angle measurements as well as the extent of spray retention improvements. This ensures that the abstract provides a comprehensive summary of both the methods used and the key findings from our study.

Comment 05:

The manuscript would benefit from incorporating more recent references, particularly in the introduction. Currently, it cites only one reference from 2024 and none from 2025, which is unusual for a paper submitted in late 2025 .

In the revised manuscript, we have incorporated additional up-to-date sources, including publications from 2024 and 2025, to strengthen the context and ensure that references reflect the latest developments in the field.

Comment 06:

The statistical analyses performed in this study appear to be basic and lack sufficient depth. There are no comparisons of means, suchANOVA , to statistically validate the differences observed between the various experimental conditions.

We thank the Reviewer for this valuable comment and the opportunity to improve and clarify our approach. In the revised manuscript, we have used statistical validation. Specifically, we applied one-way ANOVA to evaluate differences among the experimental conditions. When ANOVA indicated a significant effect (p < 0.05), pairwise comparisons were performed using Tukey’s HSD test to control for multiple comparisons and ensure robust interpretation of the data. The statistical methods are now reported at the end of Section 4.7. The analysis confirms that the observed differences between microparticles and nanoparticles on PMMA plates, including the effect of the rinsing steps, are statistically significant and support the conclusions drawn in the manuscript. Coverage differences remain significant when measured on laurel leaves, although the rinsing procedure did not produce a statistically relevant effect in this case. These comments are now included in the revised version of the manuscript.

Comment 07:

The figure and table legends should be more self-contained. Although the information may be presented in the body of the text, it is crucial that all relevant details are included in the legends for clarity and ease of understanding. This includes the number of repetitions in the experiments, the nature of the error values (standard deviation or standard error.....), and explanations of any abbreviations used within the figures or tables .

In the revised manuscript, we have ensured that all legends are more self-contained, as suggested by the Reviewer. These changes improve clarity and ease of understanding without requiring readers to refer back to the main text.

Comment 08:

Please, make sure that all references have a corresponding citation within the text and vice versa. Please, double-check the spelling of the author’s names and dates and make sure they are correct and consistent with the citations. Please, spell out all journal titles in the references section. Please, make sure that all figures and tables are cited within the text and that they are cited in consecutive order. Please, spell al the abbreviations the first time when they are mentioned in the text.

We appreciate the Reviewer’s detailed recommendations. We have corrected references, citations, author details, figure/table order, and defined all abbreviations at first mention.

Concerning the use of abbreviated journal titles in the references we have carefully followed the reference formatting instructions provided by the journal (as outlined https://www.mdpi.com/journal/plants/instructions#references) and have verified our approach by consulting already published manuscripts in Plants, where abbreviated journal titles are consistently used. Therefore, we have maintained the use of abbreviated journal titles in accordance with the journal’s guidelines.

Comment 09:

I am sorry, but the article is far from meeting the standards of the Plants journal. The manuscript requires significant revisions, particularly in terms of experimental design, statistical analysis, and the inclusion of more recent references.

We appreciate the Reviewer’s candid feedback. In the revised manuscript, we have made substantial improvements to address all concerns raised. We are confident that the revised version now meets the standards of Plants.

Reviewer 3 Report

Comments and Suggestions for Authors

It is important study for agriculture, but there some issues, which need to be taken into account:

1.Abstract contain an introductive part, which can be reduced and some numbers need to be added in it.

  1. In the introduction is should be presented why is important to study the effect of calcium carbonate particles, what is their importance
  2. Introduction and discussion section should also present what is the advantage of calcium carbonate particles compared to other particles used to retain water.
  3. Why experiments were performed on harvested leaves and not on the leaves on the growing plants?
  4. Is this first study performed with calcium carbonate particles? If no, in the discussion section these results need to be mentioned.
  5. There is no information about statistical treatment of the data
  6. Since study was performed on model systems its limitations need to be indicated in real agricultural conditions the results can be absolutely different.
  7. what about toxicity of calcium carbonate particles?

Author Response

It is important study for agriculture, but there some issues, which need to be taken into account:

  1. Abstract contain an introductive part, which can be reduced and some numbers need to be added in it.

We thank the Reviewer for this valuable suggestion. In response, we have shortened the introductive part of the abstract and included relevant numerical data to enhance its clarity and informativeness.

  1. In the introduction is should be presented why is important to study the effect of calcium carbonate particles, what is their importance

We thank the Reviewer for highlighting the need to clarify the importance of studying calcium carbonate particles. Calcium carbonate (CaCO3) is of particular interest because it is abundant in nature, widely used as a liming agent in agriculture, cost-effective, and non-toxic. These characteristics make CaCO3 a sustainable and practical choice for various agricultural applications, such as improving soil quality and potentially enhancing water retention. Its natural availability and safe profile further support its relevance for agricultural research and application. We have now emphasized these points in the revised introduction to better justify the focus of our study on calcium carbonate particles.

  1. Introduction and discussion section should also present what is the advantage of calcium carbonate particles compared to other particles used to retain water.

We appreciate the Reviewer’s suggestion regarding the advantages of using calcium carbonate particles as a wetting booster in agriculture. In the revised manuscript, we have provided a detailed explanation of why calcium carbonate is preferentially recommended for this purpose. Specifically, we discuss its abundance, cost-effectiveness, non-toxicity, and proven efficacy compared to alternative particles. These qualities make CaCO3 a sustainable and practical choice for improving soil water retention, and we have ensured that this rationale is clearly presented in the revised introduction and discussion sections.

  1. Why experiments were performed on harvested leaves and not on the leaves on the growing plants?

To reply on this legitimate question we explained in the revised manuscript (Section 4.4) why we decided to use detached leaves. The reason is basically that the trials on leaves were only the first step towards application on whole plants. Working with detached leaves allowed to follow the very same rinsing procedure as used for PMMA plates, which would be impossible in whole plant trials.

  1. Is this first study performed with calcium carbonate particles? If no, in the discussion section these results need to be mentioned.

Thanks for pointing out this unclear situation. As now mentioned in the introduction the first study has been done with CaCO3 particles.

  1. There is no information about statistical treatment of the data

We thank the Reviewer for this valuable comment, which is in line with the comment raised by Reviewer 2. In the revised manuscript we have included statistical analysis of the coating coverages. The statistical methods are reported at the end of Section 4.7. We thank the Reviewer for this valuable comment and the opportunity to improve and clarify our approach. In the revised manuscript, we have used statistical validation. Specifically, we applied one-way ANOVA to evaluate differences among the experimental conditions. When ANOVA indicated a significant effect (p < 0.05), pairwise comparisons were performed using Tukey’s HSD test to control for multiple comparisons and ensure robust interpretation of the data. The statistical methods are now reported at the end of Section 4.7. The analysis confirms that the observed differences between microparticles and nanoparticles on PMMA plates, including the effect of the rinsing steps, are statistically significant and support the conclusions drawn in the manuscript. Coverage differences remain significant when measured on laurel leaves, although the rinsing procedure did not produce a statistically relevant effect in this case. These comments are now included in the revised version of the manuscript.

  1. Since study was performed on model systems its limitations need to be indicated in real agricultural conditions the results can be absolutely different.

We appreciate the Reviewer’s attention to the limitations of our study design. In the discussion section of the revised manuscript, we have explicitly stated that the results are subject to certain limitations due to the use of model systems. We acknowledge that outcomes observed under controlled conditions may differ from those in real agricultural environments, and have highlighted this point to ensure transparency and guide future research directions.

  1. what about toxicity of calcium carbonate particles?

In the introduction of the revised manuscript we added an explanation on why CaCO3 particles are suitable for the use in agriculture. Part of the explication is that it has been proven that CaCO3 is not toxic to the studied organisms.

Round 2

Reviewer 2 Report

Comments and Suggestions for Authors

Dear Authors,

I recently had the opportunity to review your manuscript titled " Effect of CaCO3 particle size on surface wetting and adhesion: Studies on PMMA model substrates and Laurus nobilis leaves" (Manuscript Number: plants-3961364). Your study certainly presents elements of interest and novelty. After carefully reviewing the updated version, I appreciate the revisions and additions you've made.

I would like to thank you for taking my suggestions into account.

Kind regards,

Reviewer

Author Response

Comment 01:

Dear Authors,

I recently had the opportunity to review your manuscript titled " Effect of CaCO3 particle size on surface wetting and adhesion: Studies on PMMA model substrates and Laurus nobilis leaves" (Manuscript Number: plants-3961364). Your study certainly presents elements of interest and novelty. After carefully reviewing the updated version, I appreciate the revisions and additions you've made.

I would like to thank you for taking my suggestions into account.

Kind regards,

Reviewer

Thank you very much for your positive feedback and for acknowledging the revisions and additions made to our manuscript. We greatly appreciate your thoughtful comments and suggestions, which have been invaluable in strengthening the quality and clarity of our work. We are pleased that our efforts to address your recommendations have met your expectations, and we are grateful for your time and support throughout the review process.

Reviewer 3 Report

Comments and Suggestions for Authors

Authors  have improved their manuscript.

However, limitations of the study need to be mentioned in the Conclusion section.

Author Response

Comment 01:

Authors  have improved their manuscript.

Thank you for recognizing the improvements made to our manuscript. We carefully considered your previous suggestions and have worked to address each point thoroughly. Your feedback has greatly contributed to enhancing the quality and clarity of our work.

Comment 02:

However, limitations of the study need to be mentioned in the Conclusion section.

Thank you for your valuable suggestion regarding the inclusion of study limitations. We have addressed this point by explicitly mentioning the limitations in the Conclusion section, ensuring that they are clearly presented for readers. We appreciate your guidance in strengthening the transparency of our work.
